# A Plant Strategy: Irrigation, Nitrogen Fertilization, and Climatic Conditions Regulated the Carbon Allocation and Yield of Oilseed Flax in Semi-Arid Area

**DOI:** 10.3390/plants13182553

**Published:** 2024-09-11

**Authors:** Haidi Wang, Bangqing Zhao, Yuhong Gao, Bin Yan, Bing Wu, Zhengjun Cui, Yifan Wang, Ming Wen, Xingkang Ma

**Affiliations:** 1State Key Laboratory of Aridland Crop Science, Lanzhou 730070, China; wanghd1823@163.com (H.W.); bqzhao2024@163.com (B.Z.); gsau_yanbin@163.com (B.Y.); wub@gsau.edu.cn (B.W.); wangyf@gsau.edu.cn (Y.W.); wmalaer@163.com (M.W.); lingstraitor@163.com (X.M.); 2College of Agronomy, Gansu Agricultural University, Lanzhou 730070, China; 3College of Life Science and Technology, Gansu Agricultural University, Lanzhou 730070, China; 4College of Agronomy, Tarim University, Alaer 843300, China; czjungsau@163.com

**Keywords:** oilseed flax, dry matter, non-structural carbohydrates, grain yield, water use efficiency

## Abstract

The injudicious use of water and fertilizer to maximize crop yield not only leads to environmental pollution, but also causes enormous economic losses. For this reason, we investigated the effect of nitrogen (N) (N0 (0), N60 (60 kg ha^−1^), and N120 (120 kg ha^−1^)) at different irrigation levels (I0 (0), I1200 (budding 600 m^3^ ha^−1^ + kernel 600 m^3^ ha^−1^), and I1800 (budding 900 m^3^ ha^−1^ + kernel 900 m^3^ ha^−1^)) on oilseed flax in the Loess Plateau of China in 2019 and 2020. The objective was to establish appropriate irrigation and fertilizer management strategies that enhance the grain yield (GY) of oilseed flax and maximize water and N productivity. The results demonstrated that irrigation and N application and their coupling effects promoted dry matter accumulation (DMA) and non-structural carbohydrate (NSC) synthesis, and increased the GY of oilseed flax. The contents of NSC in various organs of flax were closely related to grain yield and yield components. Higher NSC in stems was conducive to increased sink capacity (effective capsule number per plant (EC) and thousand kernel weight (TKW)), and the coupling of irrigation and N affected GY by promoting NSC synthesis. Higher GY was obtained by the interaction of irrigation and N fertilizer, with the increase rate ranging from 15.84% to 35.40%. Additionally, in the increased yield of oilseed flax, 39.70–78.06%, 14.49–54.11%, and −10.6–24.93% were contributed by the application of irrigation and nitrogen and the interaction of irrigation and nitrogen (I × N), respectively. Irrigation was the main factor for increasing the GY of oilseed flax. In addition, different climatic conditions changed the contribution of irrigation and N and their interaction to yield increase in oilseed flax. Drought and low temperature induced soluble sugar (SS) and starch (ST) synthesis to resist an unfavorable environment, respectively. The structural equation model showed that the key factors to increasing the GY of oilseed flax by irrigation and nitrogen fertilization were the differential increases in DMA, EC, and TKW. The increases in EC and TKW were attributed to the promotion of DMA and NSC synthesis in oilseed flax organs by irrigation, nitrogen fertilization, and their coupling effects. The I1200N60 treatment obtained higher water use efficiency (WUE) and N partial factor productivity (NPFP) due to lower actual evapotranspiration (ETa) and lower N application rate. Therefore, the strategy of 1200 m^3^ ha^−1^ irrigation and 60 kg ha^−1^ N application is recommended for oilseed flax in semi-arid and similar areas to achieve high grain yield and efficient use of resources.

## 1. Introduction

Water and fertilizer are essential elements of agricultural production. Global agriculture depends on most surface water and 72% of groundwater for irrigation during agricultural production [1]. Agricultural production in semi-arid areas plays an important role in global agricultural production. Water deficit and low soil fertility are two main factors limiting the agricultural production in semi-arid areas [2]. Climate change has altered the spatial and temporal distribution of rainfall due to frequent abnormal weather occurrences, leading to significant fluctuations in the demand for irrigation water on an annual and growth stage basis [3,4]. Due to the climate characteristics of scarce rainfall and intense evaporation in semi-arid areas, irrigation and fertilization have become necessary technical measures for crop production [5]. Considering the temporal and spatial variability, it is very judicious to synchronize irrigation and fertilization with crop demand based on regional and crop characteristics to formulate water and fertilizer management strategies [6,7,8].

Adequate N supply is a necessary condition for crops to obtain the highest yield [9], but the oversupply of N leads to later senescence, plant lodging, inferior seeds, and low yield in crop production [10], and causes a series of problems, including eutrophication of water, soil acidification, and greenhouse gas [11,12,13]. Moreover, both excessive irrigation and N lead to low water and N use efficiency, and threaten the sustainability of agricultural ecological systems [14]. Therefore, the proper utilization of irrigation and N offers potential strategies that simultaneously improve crop production and ensure sustainable agricultural development in semi-arid areas [15]. There is an interaction between irrigation and N fertilizer, but the interaction effect changes with the change in environmental conditions [16,17]. The application of N significantly increases crop yield under low irrigation, but the role of N to yield is highly hidden under a high irrigation condition [18]. The effect of water and N on crop yield is mainly achieved through the effect on yield components, so it is particularly important to clarify the relationship between water and N consumption and yield components. However, the yield components are mutually restricted due to the competition for limited photosynthates and other nutrients. For example, an excessive amount of spike density in wheat inhibits grain filling and reduces grain weight [19], and competes with the grain number per spike [20]. Therefore, balancing yield components and clarifying the compensation among yield components are effective measures to achieve high crop yield [21].

Non-structural carbohydrates (NSCs), which consist of soluble sugar (SS) and starch (ST), not only have an important impact on plants to cope with extreme climatic conditions (such as seasonal drought and extreme rainfall) [22], but also provide a mobile carbon pool for plant growth and metabolism. The plant provides energy to its reproductive organs by reallocating NSC reserves accumulated through photosynthesis, further stabilizing and enhancing its productivity [23]. However, the limited understanding of the mechanism of carbon resource (such as NSCs) allocation between different tissues of plants (crops) limits the formation of their target traits [24,25]. The grain filling capacity of most field crops depends on the temporary storage of NSCs in their vegetative organs, and the allocation of NSCs has been identified as an important feature of the maximum yield [26]. Changes in environmental conditions can readily induce alterations in NSC levels, and analyzing these changes can provide crucial insights for revealing the adaptation of crops to a specific environment and the mechanism of yield formation [27]. The increase in NSC concentration is attributed to the application of N, which results in an increase in crop yield [25]. Maize (*Zea mays* L.) vegetative organs show an increased NSC accumulation under low N stress, enhancing the contribution of NSCs in stems to yield and subsequently reducing yield losses due to low N stress [28]. NSC concentration shows a gradient difference between source organs and sink organs in terms of temporal and spatial pattern and is susceptible to changes in climatic conditions [22,25]. Seasonal drought stress frequently results in diminished and unpredictable crop yields, particularly in semi-arid areas, but NSCs in plants are involved in drought tolerance through the interconversion between ST and SS to ensure higher yield [29,30,31]. Irrigation is an advantageous measure for crop production in semi-arid areas. The amount of irrigation highly affects the distribution of NSCs in crop vegetative organs, and leaves NSCs increased with the increase in irrigation amount [32]. Carbon allocation between the source and sink organs and the enhancement of sugar transport are responses of crops to adverse climatic conditions under drought stress [32]. Therefore, it is conducive to further unravel the mechanisms of yield formation to clarify the effects of N application and irrigation on the NSCs of crop organs and their relationship with yield and yield components under different climatic conditions.

Oilseed flax (*Linum usitatissimum* L.) is a major oilseed crop after soybean (*Glycine max* L.), oilseed rape (*Brassica campestris* L.), sunflower (*Helianthus annuus* L.), and peanut (*Arachis hypogaea* L.), which is widely cultivated in temperate regions [33,34]. Flaxseed contains various functional constituents, including lignans, lipids, proteins, and dietary fibers, and flaxseed has been used for various purposes [35,36]. Currently, oilseed flax is becoming popular for its high nutritional value and is increasingly planted in the arid and semi-arid areas of Northwest China [8]. To obtain high yield with quality, the oversupply N has become a common practice in oilseed flax production [33,37], which is not only costly but also pollutes the environment. Thus, improving the grain yield of oilseed flax through advancing water and N productivity is pivotal for obtaining quality yield in semi-arid areas. Previous studies have shown that increasing the amount and frequency of irrigation can make flax plants healthier, improve lodging resistance, and increase yield [38]. With the increase in irrigation, the water consumption of oilseed flax in a farmland increases, and the stage of water consumption indicates that anthesis to maturity is higher than budding to anthesis [39]. There are complementary effects of nitrogen application and irrigation on the grain yield of oilseed flax [40]. The coupling of irrigation and nitrogen fertilizer increases grain yield by optimizing the accumulation and distribution of dry matter, nitrogen transportation in vegetative organs after anthesis, and the contribution rate to grains to increase grain yield [40,41]. However, little is known about the carbon allocation strategies of oilseed flax and their relationship with yield and yield components under different irrigation and N application managements, and the contribution of irrigation and N application to yield increase is uncertain with climate change. Therefore, we conducted 2-year field experiments to optimize the yield of oilseed flax via the proper regulation of N and irrigation in the arid and semi-arid areas of Northwest China in 2019 and 2020. The main research objectives are as follows: (1) investigate the response of different water and fertilizer combinations on DMA, NSC, GY, and water and N productivity of oilseed flax; (2) clarify the contribution of irrigation, N, and their interaction to the increase in GY of oilseed flax; and (3) propose an appropriate irrigation and N application strategy for oilseed flax in semi-arid areas of Northwest China. The research results can provide a theoretical basis and technical reference for sustainable cultivation while synergistically utilizing water and N for oilseed flax in semi-arid areas.

## 2. Results

### 2.1. Dry Matter Accumulation under Different Treatments

Nitrogen (N) had a significant effect on the accumulation of DMA, but irrigation (I) and I × N had significant effects on the DMA of some flax growth stages in both years (Table 1). Due to irrigation at the budding stage in 2019 and 2020, there were no obvious variations in DMA at the seedling stage. During two growing seasons, the DMA of oilseed flax increased with the increase in irrigation amount (except for the kernel stage in 2020), compared with the N0, I1200, and I1800 levels, which were significantly increased from the anthesis to the maturity stage. The DMA of oilseed flax showed uncertainty with the increase in N application rate, which varied with year and growth period. Overall, DMA was significantly higher than that of N0 (except at the seedling stage in 2019), 6.45–23.22% and 5.22–31.96% higher at the N60 and N120 levels, respectively.

In summary, DMA was promoted by N application and irrigation, but did not show a consistent trend of increasing with increasing N application and irrigation amounts. There were no significant differences between low N (N60) and high N (N120) and between low irrigation (I1200) and high irrigation (I1800) in most cases.

DMA under different treatments is shown in Figure 1. The DMA of oilseed flax was highest under the I1800N1200 treatment from the anthesis stage to the maturity stage in 2019. The highest DMA was observed under the I1200N60 treatment at the kernel stage and the I1800N120 treatment at the maturity stage in 2020.

### 2.2. Soluble Sugar and Starch Content under Different Treatments

#### 2.2.1. Variance Analysis of Soluble Sugar and Starch Content in Various Organs of Oilseed Flax at Pre-Anthesis and Post-Anthesis

N had significant effects on SS and ST content of oilseed flax in all organs at pre-anthesis and post-anthesis in 2019 and 2020 (Table 2). The effect of irrigation on ST content was greater than that on SS content. There were significant effects on ST (except for leaves at pre-anthesis in 2019), and SS content was different in various organs. I × N had a significant effect on the stem SS content at pre-anthesis in 2019 and at post-anthesis in 2020, and had significant effects on the capsule ST content in both 2019 and 2020.

#### 2.2.2. Soluble Sugar and Starch Content in Leaves

In the 2019 and 2020 growing seasons, the content of SS and ST in leaves increased with the increase in irrigation and N application levels, but there was no significant difference between I1200 and I1800 and N60 and N120 levels (Figure 2). In 2019, the leaf SS content was significantly increased by 11.65% and 12.70% at the I1800 level compared with the I0 level at pre-anthesis and post-anthesis, respectively. The highest SS content was obtained at the I1800 level in 2020, but it had insignificant differences at the I0, I1200, and I1800 levels at post-anthesis. Compared with the N0 level, N120 significantly increased the leaf SS content by 9.89–22.73% in both growing seasons as a whole. The leaf ST contents under irrigation and N fertilizer treatment were significantly higher than that without irrigation and N application by 4.28–17.11% and 6.83–19.74%, respectively. The maximum leaf SS content was obtained under the I1800N120 treatment over two growing seasons, and the lowest ST content was observed under the I0N0 treatment.

#### 2.2.3. Soluble Sugar and Starch Content in Stems

Irrigation and N application increased the content of SS and ST in stems of oilseed flax, and the content of SS and ST increased with the increase in N application amount during two growing seasons, which had a significant difference among the N0, N60, and N120 levels (Figure 3). The stem SS content increased with increasing irrigation levels in 2019, and the I1800 level was 46.17% and 18.02% higher than the I0 and I1200 levels at pre-anthesis, respectively. The I1200 level resulted in the highest stem ST content at pre-anthesis and post-anthesis in 2020, which was significantly higher than the I0 and I1800 levels at post-anthesis. The stem SS content was higher under the I1800N120 treatment at pre-anthesis and post-anthesis in 2019. The highest stem SS content was obtained under the I1200N120 treatment, which was 0.10–35.90% and 13.70–72.08% higher than other treatments at pre-anthesis and post-anthesis in 2020, respectively. The stem ST content under the I0N0 treatment was significantly lower than that under other treatments, 16.768–39.39% and 5.35–47.14% at post-anthesis in 2019 and 2020, respectively.

#### 2.2.4. Soluble Sugar and Starch Content in Capsules

During two growing seasons, the content of SS and ST in capsules of oilseed flax increased with the increase in N application rates (Figure 4). N application significantly increased ST content; irrigation increased the content of SS and ST by 3.16–30.56% and 6.97–13.43% compared with I0. ST content in the I0N0 treatment was significantly lower than that in other treatments by 20.07–41.33% and 15.97–35.91% in 2019 and 2020, respectively.

#### 2.2.5. Correlation of Soluble Sugar and Starch Content with Yield

Correlation analysis revealed that the SS content in stems and leaves at pre-anthesis and post-anthesis positively correlated with EC and GY (*p* < 0.01) (Table 3). The stem ST content followed a strong positive relationship with EC, GN, and GY while presenting a negative relationship with TKW (*p* < 0.01). The leaf ST content was negatively correlated with EC and GN, but positively correlated with TKW (*p* < 0.01). There was a significant positive correlation between the capsule SS content and EC and GN, but a negative correlation with TKW (*p* < 0.01). On the contrary, the capsule ST content had a negative correlation with GN and a positive correlation with TKW, but the positive relationship was observed between the content of SS and ST in capsules and GY (*p* < 0.01).

### 2.3. Yield Performance

#### 2.3.1. Yield Components and Grain Yield

In the 2019 and 2020 growing seasons, irrigation had significant effects on EC (*p* < 0.01 in 2019, *p* < 0.05 in 2020) and had highly significant effects on GY (*p* < 0.01) (Table 4). In the 2019 growing season, irrigation, N, and their interactions had highly significant effects on TKW, but had insignificant effects on TKW in 2020. Over two growing seasons, N had significant effects on GY (*p* < 0.05). There were no significant effects of irrigation, N, and their interactions on GN.

During the 2019 growing season, EC increased with the increase in irrigation levels, and was significantly higher at the I1800 level than that at the I0 and I1200 levels, with the maximum at the I1200 level in 2020. GY increased with the increase in irrigation levels; significant differences were observed between three irrigation levels. Additionally, GY first increased and then decreased with the levels of N application, and all of them reached the maximum at the N60 level in both growing seasons. Higher GY was observed under the I1200N60, I1800N60, and I1800N120 treatments in 2019 and 2020. In 2019, TKW increased with the increase in irrigation levels, and first increased and then decreased with the increase in N application levels. The I1800N60 treatment obtained the highest TKW.

A positive relationship was observed between EC and TKW, but a negative relationship existed with GN (Figure 5), indicating that the increased GY was due to the higher EC and TKW in this study.

#### 2.3.2. Yield Increase Rate and Contribution to Yield Increase

From N0 to N60, the GY of oilseed flax increased significantly by 5.40% and 14.57% in the 2019 and 2020 growing seasons, respectively (Table 5). There was no significant difference in GY between the N60 and N120 levels in 2019. Compared with the N60 level, GY decreased significantly by 6.00% at the N120 level in 2020. In the 2019 and 2020 growing seasons, GY was significantly increased at the I1200 level by 11.31% and 13.45% over the I0 level, which increased at the I1800 level by 4.83% and 6.67%, over the I1200 level, respectively. The highest GY was obtained at the I1800 level, which was 16.69% and 21.02% higher than that at the I0 level in both growing seasons, respectively.

Table 6 shows the GY of oilseed flax’s increase rate in each treatment compared with I0N0 and the contribution rate of irrigation, N, and I × N to yield increase. Compared with I0N0, the yield increase rate of other treatments ranged from 5.63% to 16.77% due to the N application under the I0 condition. The yield increase rates of I1200N0 and I1800N0 were due to irrigation, while those of other treatments were mainly due to irrigation, N application, and I × N under the I1200 and I1800 conditions. The lowest GY was obtained under the I0N0 treatment, at 1048.44 and 960.57 kg ha^−1^ in 2019 and 2020, respectively. The GY of oilseed flax was higher under the interaction of irrigation and N; the I1800N60 treatment had the highest GY, which was significantly increased by 23.15% and 35.40% compared with the I0N0 treatment in 2019 and 2020, respectively.

The contributions of irrigation, N, and I × N to yield increase in the I1800N60 treatment were 18.74%, 73.40%, and 7.87% in the 2019 growing season, which were 30.01%, 45.05%, and 24.93% in the 2020 growing season, respectively. The contribution of irrigation to the increased yield was greater than that of nitrogen for the interaction of irrigation and N except for the I1200N60 treatment in 2020. The contributions of N and irrigation under high N conditions to the increased yield of oilseed flax in the 2020 growing season were both greater than those in 2019 growing season. The contribution of I × N to the increased yield was higher under the high N condition in 2019 but higher under low N conditions in 2020.

### 2.4. Water and Nitrogen Productivity under Different Treatments

In both growing seasons, irrigation had highly significant effects on ETa, WUE, NPFP, and W*_I_*UE (Table 7). N application had a highly significant effect on NPFP (*p* < 0.01), and N had a highly significant effect on ETa in 2019 and W*_I_*UE in 2020. Over two growing seasons, I × N had significant effects on NPFP (*p* < 0.05).

Under the same N application levels, ETa and NPFP increased significantly with the increase in irrigation levels in both growing seasons (Table 7), while W*_I_*UE decreased significantly with the increase in irrigation levels. The ETa of treatments with and without N application were significantly higher than that of other treatments at the I1800 level. Higher WUEs were observed under no irrigation treatments, which were significantly higher than the other treatments in 2020. W*_I_*UE at the I1200 level was significantly higher than that at the I1800 level, and the highest W*_I_*UE was observed under the I1200N60 treatment in the 2020 growing season, which was significantly higher than that under other treatments by 7.35–60.95%.

NPFP decreased significantly with the increase in N application in both growing seasons, which was the highest under the I1800N60 treatment, followed by the I1200N60 treatment, but the difference between them was not significant. In the 2019 growing season, ETa increased significantly under the condition of N application compared with that without N application.

In summary, irrigation had a greater effect on ETa, WUE, NPFP, and W*_I_*UE than N, and water-N coupling was conductive to increasing NPFP.

### 2.5. The Effects of Irrigation and Nitrogen on Dry Matter Accumulation at the Maturity Stage, Grain Yield, and Yield Components Based on a Structure Equation Model

The structural equation model revealed the relationship between farmland management (irrigation and N) and DMA at the maturity stage, GY, and yield components (Figure 6). In 2019, irrigation had significant direct effects on DMA at the maturity stage and GY. N had a significant direct effect on DMA, but had insignificant direct effects on GY. TKW had a positive effect on GY. In 2020, irrigation and N had positive effects on DMA, but did not directly affect GY. DMA, EC, and TKW had significant positive effects on GY. The path coefficients (λ) were 0.78, 0.24, and 0.18, respectively.

## 3. Discussions

### 3.1. Dry Matter Accumulation

Irrigation and N application are important management factors affecting crop growth and yield [42]. Appropriate water and N management can achieve the objective of promoting fertilizer with water and regulating water with fertilizer [43]. In this study, N application promoted the DMA of oilseed flax at each growth stage, and increased or decreased slightly with the increase in N application. The result can explain that appropriate nitrogen application can slow down the degradation of chlorophyll, prolong the functional period of leaves, and enhance the photosynthetic production capacity of leaves [28,44]. Some studies have shown that irrigation has a positive effect on crop DMA, and that there is a positive correlation between aboveground DMA and irrigation amount [45]. This result is in line with the present study. It is possible that the experiment was conducted in semi-arid areas, irrigation had a greater effect on the yield formation of oilseed flax, and the irrigation levels of I1200 and I1800 did not reach the threshold of the flax irrigation level. Irrigation had a significant effect on DMA at each growth stage after irrigation in 2019, but had no significant effect in 2020, probably due to drought caused by low rainfall in April–May (seedling–budding) and July (Kernel) of 2019, which was compensated by the compensation effect of rewatering after irrigation, indicating that irrigation was essential for agricultural production under drought conditions. To sum up, irrigation coupled with N produced more DMA than their sole supplementation. On the one hand, it might be explained that the application of N dramatically accelerated crop growth [46], and the faster growth would cost more water. On the other hand, soil water deficit limited the function of fertilizer [8].

### 3.2. Non-Structural Carbohydrates

SS and ST were the main substances that constituted GY, and the NSC content in vegetative organs was determined by the photosynthetic capacity of leaves [47]. The NSC content in stems and leaves of oilseed flax had an obvious reduction at post-anthesis compared with that at pre-anthesis in this study. The result may be explained by the translocation of NSC from the stems and leaves to the capsules of oilseed flax or the reduced photosynthetic capacity at post-anthesis. The regulation of fertilization and irrigation amount can promote the accumulation of NSC at pre-anthesis and post-anthesis to maximize yield [48]. Our results indicated that both irrigation and N application favored NSC content in various organs of oilseed flax, and similar results were also discovered in previous works [49,50].Both water stress and drought stress have been reported to cause faster remobilization of pre-stored carbon in vegetative organs into grains [48]. The results of 2 years’ experiment in this study showed that the ST contents in stems and leaves under no irrigation treatment were significantly lower than that under irrigation treatment, which were in line with the strategy of plants’ response to drought stress, in which they input limited NSC into the roots and consumed ST stored in various organs (leaves, stems, and roots) to satisfy the needs of plant metabolism and osmoregulation [51]. Hoch et al. [52] reported that drought stress inhibits plant growth more than C supply; the conclusion was inconsistent with that in this study. Compared with irrigation, no irrigation inhibited the growth of oilseed flax, while reducing the content of SS and ST in each organ in this study.

Some studies found that fertilization had an insignificant effect on NSC content in plants under different irrigation levels, and high N application led to higher NSC content [53,54]. The content of SS and ST in oilseed flax organs increased due to N application under different irrigation levels, but there was no significant difference between low N and high N in general in this study. The result can explain that N assimilation consumes the carbon skeleton and energy of carbohydrates when plants absorb large amounts of N under high N condition, even leading to a significant decrease in NSC content [55].

The leaf ST content of oilseed flax in 2020 was higher than that in 2019, while the stem ST content was opposite, which was caused by the high rainfall in 2020. It was possible that water deficit decreased the concentration of starch in plant leaves and increased the soluble sugar, and the reserve of NSC in stems was more important in unfavorable environments [56,57]. The capsule SS content in the 2019 growing season was greater than that in the 2020 growing season, while the stem ST content was opposite. The result can explain that a crucial physiological mechanism for crop tolerance to an unfavorable environment involved the conversion of ST and SS [28]. In addition, the capsule SS content was comparable with the ST content in the 2020 growing season and nearly twice as much as the capsule ST content in the 2019 growing season, indicating that drought may promote SS synthesis, because SS was a signal substance for plants to adapt to the environment [58]. Environmental stress induced the conversion of ST to SS in plant tissues, which regulated the osmotic potential of the cells to resist and adapt to the stressful environment; the result was similar to the changes in capsules in this study [59]. The ST content usually increased in plants at low temperatures [60,61]. The capsule ST content in the 2020 growing season was higher than that in the 2019 growing season in this study. The reason for this result may be that the temperature in July in 2020 was lower than that in the 2019 growing season, which was consistent with previous studies.

The degree of grain filling mainly depends on the sink capacity, but NSC is necessary for grain filling [48,62,63]. The storage of NSC in stems at pre-anthesis was closely related to grain filling [54]. However, higher ST in leaves and capsules was beneficial to the increase in TKW, while SS in various organs had a negative effect on TKW in this study, indicating that the main source of oilseed flax grain filling was ST in leaves and capsules. The sink, as an important recipient of NSC [64], and the transportation and distribution of NSC were through feedback from the sink [57]. The remobilization of carbon reserves accumulated in stems during the nutrient growth period contributed to the sink strength of cereals [65,66]. The sink strength included sink size and sink activity [67]. Both the contents of SS and ST in stems during vegetative growth (pre-anthesis) and reproductive growth (post-anthesis) effectively promoted EC, and ST had a significant positive effect on GN. It showed that NSC in stems was beneficial to improve the sink strength due to the increased sink size, while TKW was decreased because NSC was insufficient to cover the increased sink. The SS in leaves and capsules had positive effects on EC, but ST inhibited the formation of EC and GN and had positive effects on TKW. The result was an increase in TKW by reducing the inventory and the transfer of sufficient NSC to the grain. SS and ST had significant positive effects on GY (except for leaf ST at post-anthesis).

In conclusion, the GY of oilseed flax depended on sink size, and both of them were related to the content of NSC in various organs. The differences of NSC content in various organs of oilseed flax were caused by the differences of rainfall distribution and air temperature in different years, which further affected the sink size. The sink size was determined by EC, GN, and TKW; there was a restrictive relationship between them due to the limitation of sink size. The total effect of EC, GN, and TKW on GY showed a positive effect as an increase and a negative effect as a decrease in GY.

### 3.3. Grain Yield and Yield Components

Crop yield and yield components can be regulated through irrigation and N application [29]. Although water was the main driving factor of crop yield, a possibility was that irrigation amount increase did not lead to an increase in yield [29]. In this study, the GY of oilseed flax increased with the increase in irrigation levels, but the increase in GY decreased gradually between the I0, I1200, and I1800 levels, which indicated that the effect of increasing GY of high irrigation amount was better than that of low irrigation amount under the condition of water deficit. The effect of increasing GY was limited under higher irrigation conditions.

When the application rate of N fertilizer exceeds the tolerable range, the growth and development of crops will be seriously affected by reduced stress tolerance, excessive vegetative growth, and reduced light energy utilization, resulting in a decrease in yield [68]. In this study, the EC and GY of oilseed flax increased at first and then decreased with the increase in N application rate, which may be due to the excessive tiller stems and branches caused by fierce competition among individual plants, and finally led to lower EC. Another reason may be that excessive N application increased the unproductive tiller stems and branches, wasting more accumulated dry matter [69]. The TKW, EC, and GY of oilseed flax showed similar changes under all treatments in 2019, but were not significantly affected by irrigation and N in 2020. One possible explanation for this is that the distribution of rainfall led to water stress during the grain-filling period, reducing grain weight [70]. The GY of oilseed flax was positively correlated with EC and TKW, but negatively correlated with GN. The main reason was the competitive relationships among the yield components of oilseed flax; the negative effect of GN on yield can be compensated by EC and TKW. Although GY increased due to irrigation and N application, the effect of irrigation was greater than that of N application. This was in line with the findings of Gao et al. (2023) [8] in a similar experimental environment. Due to more rainfall from April to July 2020, the contributions of N and irrigation under high N conditions to the increased yield of oilseed flax in the 2020 growing season were both greater than those in the 2019 growing season, which can explain that N fertilizer could dramatically reduce GY loss due to water stress [71]. The contribution of I × N to the increased grain yield was higher under high N condition in 2019, but higher under low N conditions in 2020. The results showed that the effects of irrigation and N would be adjusted accordingly under different climatic conditions. Under the same irrigation conditions, the contribution of N to the increased yield decreased with increasing levels of N application, but the contribution of irrigation to the increased yield increased with increasing levels of irrigation at the same level of N application, suggesting that the role of N was constrained by irrigation, whereas irrigation was less affected by N fertilizer.

### 3.4. Water Use Efficiency and Nitrogen Partial Factor Productivity

Different irrigation and N application levels significantly affected plant water-N productivity characteristics, including ETa, WUE, W*_I_*UE, and NPFP (Table 7). WUE was higher under deficit irrigation, whereas WUE decreased under excessive irrigation but could be improved by optimizing irrigation [72]. In this study, compared with no irrigation, WUE decreased significantly under the I1200 and I1800 irrigation levels. Although the highest WUE was observed without irrigation, it was not conducive to agricultural production. This study observed results similar to that of Cui et al. [73]; irrigation was the main factor in increasing oilseed flax production in dryland, but high-yield and high-resource utilization were contradictory. Additionally, although higher GY was obtained at the I1800 level, higher WUE and W*_I_*UE were observed at the I1200 level.

Under all treatments, the WUE of oilseed flax in 2019 was higher than that in 2020, and the ETa in 2020 was higher than that in 2019. The reason for this result was that the 2020 growing season provided more rainfall for the farmland system, and irrigation amount may exceed the maximum plant water demand. Thus, irrigation level increased from I1200 to I1800, resulting in a slight increase in GY but an increase in ineffective evapotranspiration, thus reducing WUE. However, rainfall was deficient, leading to soil becoming dry in the 2019 growing season. As a result, more irrigation water was needed to moisten the root zone soil and promote crop water uptake and production [74]. Mounkaila Hamani et al. [75] observed that increased N application rates within a certain range enhanced crop WUE. However, N application had no significant effect on WUE in our study, and W*_I_*UE at the I1200 level was significantly higher than that at the I1800 level, mainly due to the increase in yield being less than the increase in irrigation amount. It was reported that low irrigation and high N were beneficial to improve WUE, and high irrigation and low N to improve NPFP [76]. In the present study, both NPFP and yield decreased significantly with increasing N application levels. The reason for this result may be that the excess nitrogen input exceeds the maximum nitrogen requirement of the plant, which leads to higher nitrogen losses [77,78]. Meanwhile, NPFP increased significantly with the increase in irrigation levels, which may lead to the decrease in plant nitrogen utilization capacity, which was related to the decrease in soil nitrogen availability or root nitrogen absorption capacity under water deficit. Sufficient water promotes efficient nitrogen utilization by plants [79].

Other studies hold a different view from this study that although irrigation promoted fertilizer absorption, excessive irrigation resulted in fertilizer leaching outside the root zone, which was not conducive to nitrogen uptake by plant roots, and therefore, NPFP increased and then decreased with the increase in irrigation [8]. In addition, we found that the interaction between irrigation and N had a significant effect on NPFP. NPFP was the highest under the I1800N60 treatment, followed by the I1200N60 treatment, but there was no significant difference between them. The result showed that the coupling of water and N was conducive to improving NPFP and promoting the efficient use of N fertilizer under the premise of conserving water resources and ensuring the sustainable development of agriculture.

The frequency of irrigation and fertilization should be noted as the key factors affecting crop DMA, GY, and water-N productivity when the total amounts of irrigation and N application were constant in addition to irrigation and N application [80]. Evaluating the frequency of irrigation and fertilization was beneficial to the optimization of the water and fertilizer management strategy.

## 4. Materials and Methods

### 4.1. Site Description

Field experiments were conducted in 2019–2020 at the Dingxi Academy of Agricultural Science (34.26° N, 103.52° E and altitude 2040 m), Gansu Province, China. The average annual rainfall in the study area is approximately 380 mm, with an average annual temperature of 6.3 °C and an annual sunshine duration of 2453 h; the frost-free period is about 140 days. The soil of the experimental site is loess, which was terraced. Before planting oilseed, soil samples from a 0 to 30 cm soil layer were collected to determine their basic soil physicochemical properties. The basic data of the soil are organic matter, 17.51 g kg^−1^; total nitrogen, 1.00 g kg^−1^; available nitrogen, 56.50 mg kg^−1^; total phosphorus, 0.85 g kg^−1^; available phosphorus, 26.43 mg kg^−1^ (NaHCO_3_ extraction–Mo–Sb anti spectrophotometric method); available potassium, 108.37 mg kg^−1^ (NH_4_OAc extraction–flame photometric method); and pH 8.3. The rainfall and temperature of the growth seasons are as follows (Figure 7):

The blue bar represents the total rainfall in a month, and the average rainfall represents the monthly rainfall averaged from the total rainfall from April to August.

### 4.2. Experimental Design and Field Management

The experiments were arranged in a two-factor split-plot design. In the experiment, there were three irrigation levels (I_0_ (0 m^3^ ha^−1^), I_1200_ (budding 600 m^3^ ha^−1^ + kernel 600 m^3^ ha^−1^), and I_1800_ (budding 900 m^3^ ha^−1^ + kernel 900 m^3^ ha^−1^)) as the main plot and three nitrogen levels (N_0_ (0 kg ha^−1^), N_60_ (60 kg ha^−1^), and N_120_ (120 kg ha^−1^)) as the subplot. To ensure even irrigation of each plot and minimize the edge effect, 50 cm width ridges were set up between each replicate and 40 cm width ridges were set up between each plot. Urea (N content, 46%) was selected as the nitrogen fertilizer. Each treatment was arranged with three replicates. The irrigation method was flood irrigation; irrigation water was introduced into each residential area through pipelines with a diameter of 8 cm and measured by the water meters. Two-thirds of N was applied as base fertilizer at a depth of 0–20 cm, and 1/3 of N was broadcasted on the soil surface before irrigation at the kernel stage. The oilseed flax plant was sown at a rate of 7.5 million plants ha^−1^ in 20 cm apart rows in a 10 m^2^ (2 m × 5 m) plot.

Groundwater was used for oilseed flax irrigation during the 2-year experiment. According to the public announcement of routine monitoring of water quality in centralized drinking water sources in county-level cities on the official website of the Dingxi Municipal People’s Government, the quality information of oilseed flax irrigation water in the past 2 years can be obtained (Table 8).

### 4.3. Date Collection

Oilseed flax was sown on 7 April and 5 April and harvested on 8 August 2019 and 12 August 2020 in the 1st and 2nd year of the trail, respectively. To minimize disturbance of the experimental plots by adjacent plots due to the movement of water and nitrogen, plant samples were collected to avoid being collected at the edges of the plots.

#### 4.3.1. Dry Matter Accumulation

Ten oilseed flax plants were randomly selected from each oilseed flax plot, cutting the aboveground parts to measure aboveground DMA at the seedling, budding, anthesis, kernel, and maturity stages. Fresh plant samples were immediately put in an oven at 105 °C for 30 min and then dried at 75 °C to a constant weight.

#### 4.3.2. Soluble Sugar and Starch

Ten plant samples with the same growth were randomly selected at pre-anthesis and post-anthesis, and the fresh plant samples were divided into stems and leaves at pre-anthesis and stems, leaves, and capsules at post-anthesis. The content of SS and ST in fresh plants was determined by concentrated sulfuric acid-anthracenone [25].

#### 4.3.3. Grain Yield and Yield Composition of Oilseed Flax

Approximately 15 mature plants were randomly selected from each plot to measure EC, grain number per capsule (GN), and TKW. The oilseed flax in each plot was harvested individually, and GY was measured after air-drying and selection.

#### 4.3.4. The Contribution to Yield Increase of Oilseed Flax

$$N\;(I_{i}N_{j})\;(\mathrm{Contribution}\;\mathrm{to}\;\mathrm{yield}\;\mathrm{increase}\;(\%))=\frac{\frac{GY(IiNj)-GY(IiN0)}{GY(IiN0)}}{Yield\;increase\;rate\;(IiNj)}\times 100$$$$I\;(I_{i}N_{j})\;(\mathrm{Contribution}\;\mathrm{to}\;\mathrm{yield}\;\mathrm{increase}\;(\%))=\frac{\frac{GY(IiNj)-GY(I0Nj)}{GY(I0Nj)}}{Yield\;increase\;rate\;(IiNj)}\times 100$$
I_i_ × N_j_ (Contribution to yield increase (%)) = 1 − N (I_i_N_j_) (Contribution to yield increase (%)) − I (I_i_N_j_) (Contribution to yield increase (%))
where i is the irrigation level, and j is the nitrogen application level.

#### 4.3.5. Water and Nitrogen Productivity

Soil samples were collected at depths of 0–20, 20–40, 40–60, 60–80, 80–100, 100–120, 120–140, and 140–160 cm layers before sowing and after harvesting. Each sample was placed in an aluminum specimen box and dried to constant weight at 105 °C. Soil water content (SWC) and soil water storage amount (SWS) were calculated according to the following Equations (1) and (2), respectively:(1)$$SWC\left(\%\right)=\left(FW-DW\right)/DW\times 100\;$$
(2)$$SWS\left(mm\right)=\sum SWCi\times Hi\times Di$$
where FW, DW, Hi, and Di are the fresh weight (g), dry weight (g), depth (cm), and bulk density (g cm^−3^) of soil samples. Bulk density was measured through a cutting ring.

Eta, WUE, and W*_I_*UE were calculated according to the following Equations (3)–(5):(3)ETa=P+I+∆W−R−D
(4) WUE=GY/ET 
(5)$${\;W}_{I}UE=GY/I\;$$
where WUE (kg ha^−1^ mm^−1^) is the water use efficiency, W*_I_*UE (kg ha^−1^ mm^−1^) is the irrigation water use efficiency, and ETa is the actual evapotranspiration (mm). P, I, and ∆W are the rainfall (mm), the irrigation (mm), and the change in soil water storage amount (mm) during the growth stage. R is the water runoff, and D is the drainage, which were negligible.

#### 4.3.6. Nitrogen Partial Factor Productivity (NPFP)

Nitrogen partial factor productivity (NPFP) was calculated by using the following Equation (6):(6)NPFP=GY/N rate
where GY is the grain yield (kg ha^−1^) and N rate is the nitrogen application rate (kg ha^−1^) [81].

### 4.4. Statistical Analysis

Data preprocessing was carried out using Excel 2016. All data were subjected to ANOVA using SPSS 26.0 with replication as a random effect and irrigation and N fertilizer rates as fixed effects, and differences between the means of three replicates were evaluated using least significant difference (LSD) multiple comparison tests at the *p* = 0.05 level. The figures were plotted using Origin 2021 (Systat Software Inc., San Jose, CA, USA). Due to significant influences between years by treatment interactions for most of the variables evaluated in this study, the treatment effects were assessed for each year separately. All significant differences were declared at the 0.05 probability level. A structure equation model was used to model the relationships among dry matter accumulation at the maturity stage, grain yield, and yield components of oilseed flax under different patterns of irrigation and N fertilization.

## 5. Conclusions

In this region, irrigation and nitrogen application can regulate EC and TKW by promoting SS and ST synthesis in oilseed flax, further increasing GY, but this positive impact was limited with increasing irrigation and N application amount. In addition, drought and low temperature induced SS and ST synthesis to resist an unfavorable environment, respectively. Water deficit is the main factor limiting the yield increase in oilseed flax in semi-arid areas. The contribution of N to increased GY was restricted by irrigation, while the contribution of irrigation to increased GY was less affected by N. An irrigation application amount of 1200 m^3^ ha^−1^ combined with a N fertilization rate of 60 kg ha^−1^ obtained higher grain yield and NPFP, as well as the highest W*_I_*UE, which was a suitable field management for oilseed flax production in semi-arid areas and similar areas.

## Figures and Tables

**Figure 1 plants-13-02553-f001:**
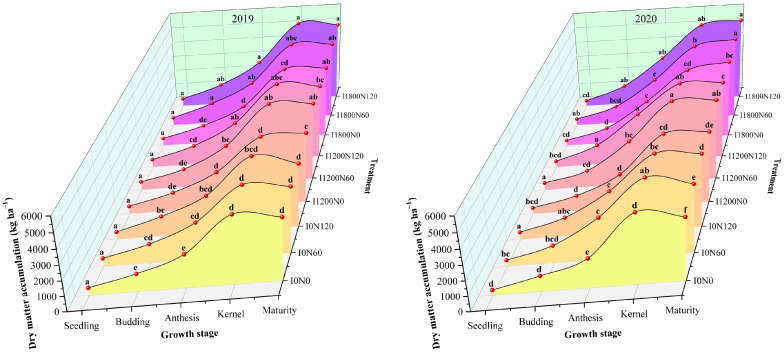
Effects of irrigation and nitrogen treatments on dry matter accumulation at different growth stages of oilseed flax in 2019 and 2020. Each value represents the mean of three replicates, and the different letters within a subgraph indicate a significant difference at *p* < 0.05. Different colors correspond to the trend of dry matter accumulation under different treatments, respectively.

**Figure 2 plants-13-02553-f002:**
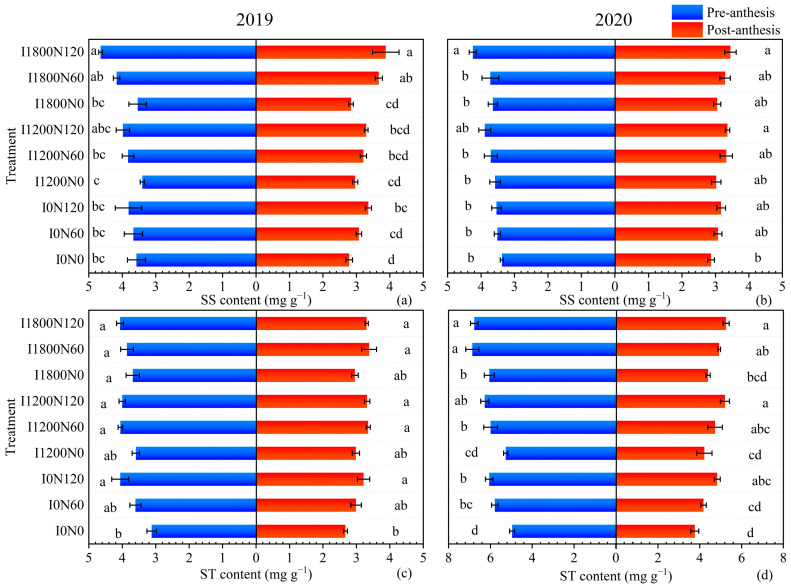
Effects of different treatments on leaves soluble sugar (**a**) and starch (**c**) content at pre-anthesis and post-anthesis of oilseed flax in 2019, and leaves soluble sugar (**b**) and starch (**d**) content in 2020. Each value represents the mean of three replicates, and the different letters within a subgraph indicate a significant difference at *p* < 0.05. Bars represent standard errors.

**Figure 3 plants-13-02553-f003:**
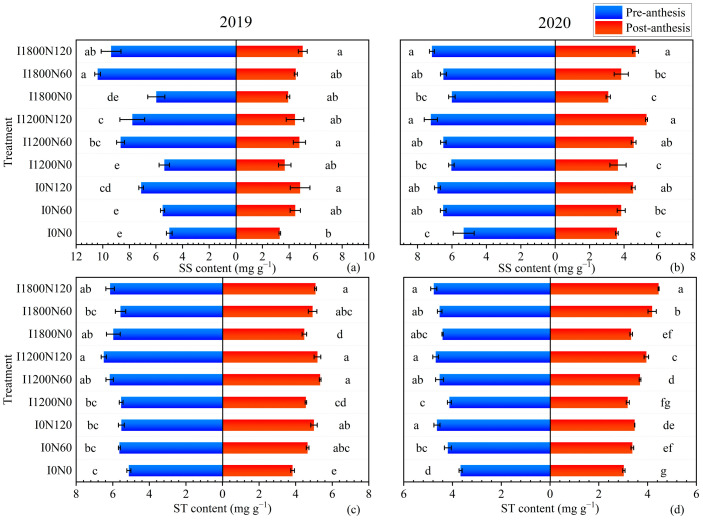
Effects of different treatments on stems soluble sugar (**a**) and starch (**c**) content at pre-anthesis and post-anthesis of oilseed flax in 2019, and stems soluble sugar (**b**) and starch (**d**) content in 2020. Each value represents the mean of three replicates, and the different letters within a subgraph indicate a significant difference at *p* < 0.05. Bars represent standard errors.

**Figure 4 plants-13-02553-f004:**
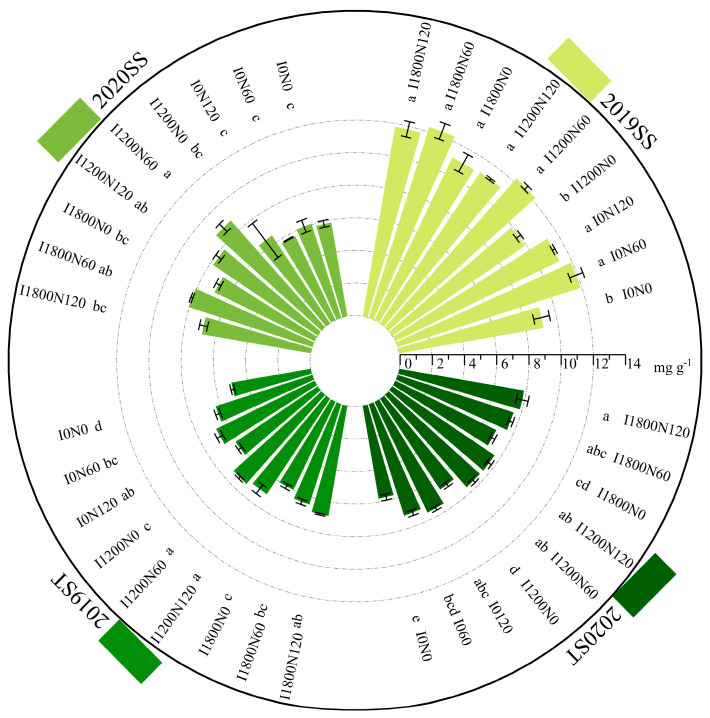
Soluble sugar and starch content in capsules measured during the kernel stage in response to different treatments in 2019 and 2020. Each value represents the mean of three replicates, and the different letters within a subgraph indicate a significant difference at *p* < 0.05. Bars represent standard errors.

**Figure 5 plants-13-02553-f005:**
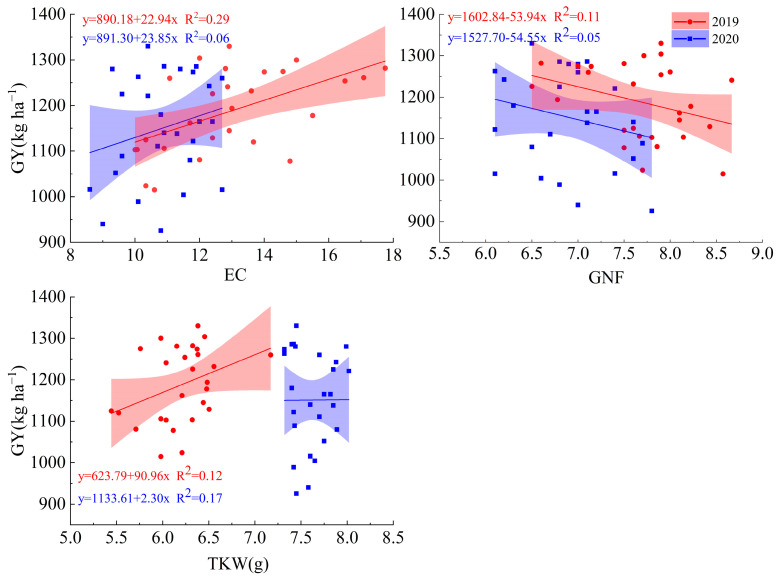
The relationships between grain yield and yield components in the 2019 and 2020 growing seasons. EC, effective capsule number per plant; GN, grain number per capsule; TKW, thousand kernel weight; and GY, grain yield. The confidence interval is 95%.

**Figure 6 plants-13-02553-f006:**
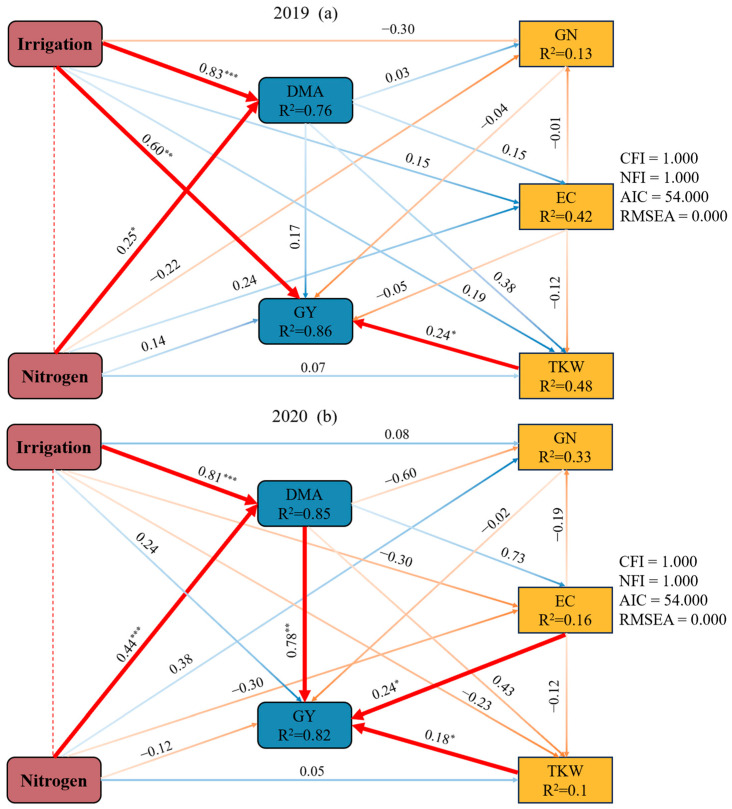
The structural equation model represents the linear relationships among irrigation amount, nitrogen application rate, GN, EC, TKW, and GY in the (**a**) 2019 and (**b**) 2020 growing seasons. EC, effective capsule number per plant; GN, grain number per capsule; TKW, thousand kernel weight; and GY, grain yield. The red arrow indicates a significant positive correlation. The blue arrow indicates an insignificant positive relationship. The orange arrow indicates an insignificant negative relationship. The numbers at the arrows are standardized path coefficients. R^2^ values represent the proportion of variance explained for each variable. The symbols ***, **, and * indicate the significance level at *p* < 0.001, 0.01, and 0.05, respectively.

**Figure 7 plants-13-02553-f007:**
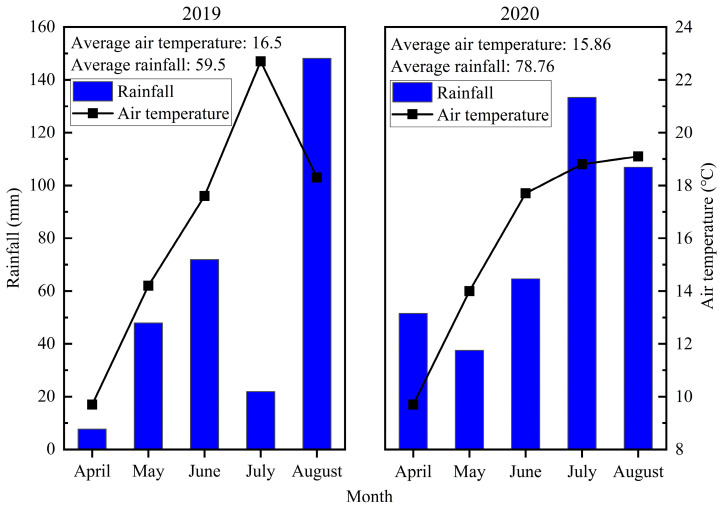
The rainfall and temperature in the growth seasons of oilseed flax.

**Table 1 plants-13-02553-t001:** Dry matter accumulation of oilseed flax under different N application and irrigation levels at different growth stages (kg ha^−2^).

Growth Stage		2019					2020				
Growth Stage		Seedling	Budding	Anthesis	Kernel	Maturity	Seedling	Budding	Anthesis	Kernel	Maturity
Nitrogen	N0	504.48 b	1220.50 b	2345.71 b	4612.74 b	4473.88 a	371.40 b	1049.21 c	2327.29 c	4697.71 c	4643.31 b
	N60	542.48 a	1299.38 a	2579.98 a	5007.56 a	4803.13 a	443.97 a	1154.82 b	2867.69 b	5179.61 a	5049.63 a
	N120	510.99 b	1333.75 a	2663.09 a	5117.65 a	4707.42 a	417.54 a	1297.71 a	3071.07 a	5153.45 b	5098.69 a
Irrigation	I0	515.06 a	1263.13 b	2390.54 b	4668.92 b	4210.73 b	413.10 ab	1155.51 a	2511.67 b	4914.18 b	4477.33 c
	I1200	516.50 a	1252.50 b	2568.71 a	4957.55 a	4816.59 a	426.63 a	1166.18 a	2826.38 a	5097.24 a	5006.27 b
	I1800	526.38 a	1338.00 a	2629.53 a	5111.48 a	4957.11 a	393.18 b	1180.06 a	2928.00 a	5019.35 ab	5308.03 a
Irrigation (I)		ns	**	**	**	**	ns	ns	**	ns	**
Nitrogen (N)		*	**	**	**	**	**	**	**	**	**
I × N		ns	*	ns	ns	ns	**	ns	**	ns	**

“*” and “**” mean significant difference at the *p* < 0.05 and *p* < 0.01 levels, respectively. “ns” means non-significant difference at the *p* < 0.05 level. Different letters in the same column means the significant difference between treatments at the *p* < 0.05 level.

**Table 2 plants-13-02553-t002:** Combined analysis of variance for SS and ST content in various organs of oilseed flax at pre-anthesis and post-anthesis in 2019 and 2020.

			2019			2020		
			Irrigation (I)	Nitrogen (N)	I × N	Irrigation (I)	Nitrogen (N)	I × N
SS	Pre-A	Stems	**	**	*	ns	**	ns
		Leaves	ns	**	ns	*	*	ns
	Post-A	Stems	ns	*	ns	*	**	ns
		Leaves	*	**	ns	ns	*	ns
		Capsules	*	**	ns	**	*	ns
ST	Pre-A	Stems	**	*	ns	**	**	ns
		Leaves	ns	**	ns	**	**	ns
	Post-A	Stems	**	**	ns	**	**	**
		Leaves	*	**	ns	**	**	ns
		Capsules	**	**	*	**	**	**

“*” and “**” mean significant difference at the *p* < 0.05 and *p* < 0.01 levels, respectively. “ns” means non-significant difference at the *p* < 0.05 level.

**Table 3 plants-13-02553-t003:** Relationship of SS and ST content between yield components.

Item			EC	GN	TKW	GY
SS	Pre-A	Stems	0.597 **	−0.077	−0.112	0.517 **
		Leaves	0.440 **	−0.076	−0.120	0.429 **
	Post-A	Stems	0.287 *	0.051	−0.019	0.282 *
		Leaves	0.523 **	−0.071	0.014	0.411 **
		Capsules	0.597 **	0.450 **	−0.758 **	0.457 **
ST	Pre-A	Stems	0.545 **	0.460 **	−0.739 **	0.405 **
		Leaves	−0.372 **	−0.643 **	0.903 **	0.107
	Post-A	Stems	0.583 **	0.325 *	−0.621 **	0.531 **
		Leaves	−0.278 *	−0.602 **	0.850 **	0.084
		Capsules	−0.050	−0.468 **	0.575 **	0.387 **

“*” and “**” mean significant difference at the *p* < 0.05 and *p* < 0.01 levels, respectively. “ns” means non-significant difference at the *p* < 0.05 level. EC, effective capsule number per plant; GN, grain number per capsule; TKW, thousand kernel weight; GY, grain yield.

**Table 4 plants-13-02553-t004:** The yield components and grain yield of oilseed flax under different treatments.

	2019				2020			
Treatment	EC (Number Plant^−1^)	GN (Number Capsule^−1^)	TKW (g)	GY (kg ha^−1^)	EC (Number Plant^−1^)	GN (Number Capsule^−1^)	TKW (g)	GY (kg ha^−1^)
I0N0	10.61 c	7.97 a	5.56 e	1048.33 e	9.47 b	7.40 a	7.54 a	960.57 e
I0N60	12.42 bc	8.02 a	6.31 abc	1103.50 de	10.77 ab	7.17 ab	7.65 a	1121.67 bc
I0N120	12.00 c	7.65 a	5.99 d	1108.67 de	10.33 ab	7.00 ab	7.61 a	1015.10 de
I1200N0	11.52 c	8.19 a	6.10 bcd	1168.6 cd	11.77 a	6.73 ab	7.56 a	1078.43 cd
I1200N60	12.93 abc	7.20 a	6.36 a	1246.33 ab	12.1 a	6.70 ab	7.70 a	1260.90 a
I1200N120	12.64 bc	7.60 a	6.06 cd	1214.33 bc	11.37 ab	7.23 ab	7.43 a	1174.77 b
I1800N0	13.52 abc	7.49 a	6.34 ab	1237.33 abc	10.73 ab	6.40 b	7.36 a	1175.67 b
I1800N60	15.92 a	7.50 a	6.46 a	1291.00 a	11.07 ab	6.80 ab	7.87 a	1300.57 a
I1800N120	15.36 ab	7.54 a	6.41 a	1276.33 ab	10.40 ab	6.70 ab	7.86 a	1272.23 a
I	**	ns	**	**	*	ns	ns	**
N	ns	ns	**	*	ns	ns	ns	**
I × N	ns	ns	**	ns	ns	ns	ns	ns

“*” and “**” mean significant difference at the *p* < 0.05 and *p* < 0.01 levels, respectively. “ns” means non-significant difference at the *p* < 0.05 level. Different letters in the same column mean the significant difference between treatments at the *p* < 0.05 level. EC, effective capsule number per plant; GN, grain number per capsule; TKW, thousand kernel weight; GY, grain yield.

**Table 5 plants-13-02553-t005:** Effects of different nitrogen and irrigation applications on the grain yield (GY) of oilseed flax.

Treatment			2019	Changes (%)	2020	Changes (%)
			GY (kg ha^−1^)		GY (kg ha^−1^)	
Nitrogen		N0	1151.44 b	–	1071.56 c	–
		N60	1213.61 a	5.40	1227.71 a	14.57
		N120	1199.78 a	–	1154.03 b	-6.00
Irrigation		I0	1086.83 c	–	1032.44 c	–
		I1200	1209.78 b	11.31	1171.37 b	13.45
		I1800	1268.22 a	4.83	1249.49 a	6.67
Increase in percentage from I0 to I1800				16.69		21.02

Changes indicating the increase in percentage of the current nitrogen/irrigation application level from the previous level. Different letters in the same column mean the significant difference between treatments at the *p* < 0.05 level.

**Table 6 plants-13-02553-t006:** Effects of different nitrogen and irrigation applications on grain yield (GY), yield increase rate, and contribution to yield increase in oilseed flax.

Treatment	2019					2020				
	GY (kg ha^−1^)	Yield Increase Rate (%)	Contribution to Yield Increase (%)			GY (kg ha^−1^)	Yield Increase Rate (%)	Contribution to Yield Increase (%)		
			N	I	I × N			N	I	I × N
I0N0	1048.33 d	–	–	–	–	960.57 d	–	–	–	–
I0N60	1103.50 cd	5.63	100.00	–	–	1121.67 bc	16.77	100.00	–	–
I0N120	1108.67 cd	5.76	100.00	–	–	1015.10 d	5.68	100.00	–	–
I1200N0	1168.67 bc	11.48	–	100.00	–	1078.43 cd	12.27	–	100.00	–
I1200N60	1246.33 ab	18.89	35.18	68.52	-3.70	1260.90 a	31.27	54.11	39.70	6.20
I1200N120	1214.33 ab	15.84	24.67	60.17	15.17	1174.77 b	22.30	40.06	70.54	-10.60
I1800N0	1237.33 ab	18.03	–	100.00	–	1175.67 b	22.39	–	100.00	–
I1800N60	1291.00 a	23.15	18.74	73.40	7.87	1300.57 a	35.40	30.01	45.05	24.93
I1800N120	1276.33 a	21.75	14.49	69.53	15.98	1272.23 a	32.45	25.31	78.06	−3.37

Yield increase rate (%) indicating the increase in the percentage of the current treatment over the I_0_N_0_ treatment. Contribution to yield increase (%) indicating the contribution of each factor (irrigation (I), nitrogen (N), and interaction of irrigation and nitrogen (I × N)) to yield increase. Different letters in the same column mean the significant difference between treatments at the *p* < 0.05 level.

**Table 7 plants-13-02553-t007:** Effects of different nitrogen and irrigation treatments on actual evapotranspiration, water use efficiency, and N partial factor productivity of oilseed flax in the 2019 and 2020 growing seasons.

	2019				2020			
	ETa (mm)	WUE (kg ha^−1^ mm^−1^)	NPFP (kg kg^−1^)	W*_I_*UE (kg ha^−1^ mm^−1^)	ETa (mm)	WUE (kg ha^−1^ mm^−1^)	NPFP (kg kg^−1^)	W*_I_*UE (kg ha^−1^ mm^−1^)
I0N0	266.42 d	3.94 a	—	—	298.72 d	3.24 b	—	—
I0N60	284.60 d	3.88 a	18.39 b	—	302.78 d	3.73 a	18.69 b	—
I0N120	273.15 d	4.06 a	9.24 d	—	296.48 d	3.44 ab	8.46 e	—
I1200N0	318.57 c	3.67 ab	—	9.74 a	419.04 c	2.59 c	—	8.99 c
I1200N60	336.48 c	3.71 ab	20.77 a	10.39 a	460.79 bc	2.75 c	21.02 a	10.51 a
I1200N120	328.00 c	3.71 ab	10.12 c	10.12 a	442.58 bc	2.68 c	9.79 d	9.79 b
I1800N0	360.57 b	3.44 b	—	6.87 b	487.01 ab	2.42 c		6.53 e
I1800N60	375.07 ab	3.44 b	21.52 a	7.17 b	528.13 a	2.47 c	21.68 a	7.23 d
I1800N120	381.11 a	3.35 b	10.64 c	7.09 b	503.49 ab	2.54 c	10.60 c	7.07 d
Irrigation (I)	**	**	**	**	**	**	**	**
Nitrogen (N)	*	ns	**	ns	ns	ns	**	**
I × N	ns	ns	*	ns	ns	ns	*	ns

“*” and “**” mean significant difference at the *p* < 0.05 and *p* < 0.01 levels, respectively. “ns” means non-significant difference at the *p* < 0.05 level. Different letters in the same column mean the significant difference between treatments at the *p* < 0.05 level. ETa, actual evapotranspiration; WUE, water use efficiency; NPFP, N partial factor productivity; and W*_I_*UE, irrigation water use efficiency.

**Table 8 plants-13-02553-t008:** Information on the quality of oilseed flax irrigation water.

Year	pH	Sulfate	Chloride	Sulfide	Nitrate	Nitrite	Ammonia Nitrogen	Sodium
2019	7.82	486.00	419.00	0.005	19.05	0.005	0.04	180.50
2020	8.21	392.00	330.00	0.005	19.6	0.005	0.04	174.00

Units are mg L^−1^; pH is dimensionless.

## Data Availability

All data included in this study are available upon request by contact with the first author.

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
