# Peer review of "A Plant Strategy: Irrigation, Nitrogen Fertilization, and Climatic Conditions Regulated the Carbon Allocation and Yield of Oilseed Flax in Semi-Arid Area"

_plants, 2024, doi:10.3390/plants13182553_

Round 1
Reviewer 1 Report
Comments and Suggestions for Authors
See pdf.

Author Response
Dear Editors and Reviewers:
Thank you for your letter and for the reviewers’ comments concerning our manuscript entitled “Irrigation, nitrogen fertilization and climatic conditions mediated the carbon allocation strategy and yield of oilseed flax in dryland” (ID: 3183177). Those comments are all valuable and very helpful for revising and improving our paper, as well as the important guiding significance to our researches. We have studied comments carefully and have made correction which we hope meet with approval. We have re-submitted our manuscript with tracked changes to highlight the revisions. The main corrections in the paper and the responds to the reviewer’s comments are as flowing:
- The title seems a bit complicated; are terms mediated and strategy necessary? It is a simple, common field experiment, is it possible to generalize the findings as “a plant (genotype) strategy” (see line 30 or 609-11)?
Response: According to the reviewer's comments, the author modified the title to "A plant strategy: irrigation, nitrogen fertilizer and climatic conditions regulated the carbon distribution and yield of oilseed flax in semi-arid areas".
Regarding the reviewer's question, "is it possible to generalize the findings as "a plant (genotype) strategy?". The authors are very grateful to the reviewer for the suggestion. The authors suggest that summarizing the results of this study as a plant (genotype) strategy may involve a wide range. Only one test variety of oilseed flax was included in this experiment, and the experiment was conducted in one location, which may not fully explain the genotype strategy of oilseed flax.
Abstract
- Line 16 Use standard terms for the stages or BBCH and the like, not every reader will understand what kernel means, here.
Response: Thanks to the reviewers for their valuable suggestions and comments.
Generally speaking, separated the entire growth stage of the oilseed flax into five stages: Seedling, Budding, Anthesis, Kernel and Mature stages. Kernel stage is widely used to indicate that after anthesis stage of oilseed flax, the seeds are gradually enriched into white, and the peel is green, that is, the previous stage of oilseed flax maturity stage.
For example, it is mentioned in the following literature on oilseed flax:
Yan, B., Wu, B., Gao, Y., et al., 2018. Effects of nitrogen and phosphorus on the regulation of nonstructural carbohydrate accumulation, translocation and the yield formation of oilseed flax. Field crops research, 219, 229-241.
Ma, X.K., Gao, Y.H., Wu, B., et al. 2023. Organic Manure significantly promotes the growth of oilseed flax and improves its grain yield in dry areas of the Loess Plateau of China. Agronomy, 13(9), 2304.
Xu, Q., Guo, L.Z., Liu, Y., Gao, Y.H. 2022. Effects of potassium and silicon fertilization on lignin metabolism and lodging resistance of oil flax stem. Chinese Journal of Eco-Agriculture, 30(9): 1451−1463.
Gao, Y.H., Wu, B., Niu, J.Y., et al. 2018. Effects of different mulching modes on soil nitrate concentration and grain yield of Linum usitatissimum in dry land. Chinese Journal of Applied Ecology, 29(10): 3283-3292.
- 17 I suggest using past tense (Pay attention to the correct use of the past and present tenses; in other places in the text the present tense would be more appropriate (e.g. 160 etc.)
Response: The author checked the full text and paid attention to the correct use of the past and present tenses. The modifications are as follows:
L17, "The objective is to establish an appropriate irrigation and fertilizer management strategies that enhance grain yield (GY) of oilseed flax," modified to "The objective was to establish an appropriate irrigation and fertilizer management strategies that enhance grain yield (GY) of oilseed flax,".
L167, "The DMA under different treatments were shown in Fig. 1." modified to "The DMA under different treatments are shown in Fig. 1".
L629, "where, FW, DW, Hi, and Di were the fresh weight (g), dry weight (g), depth (cm) and bulk density (g cm-3) of soil samples." modified to "where, FW, DW, Hi, and Di are the fresh weight (g), dry weight (g), depth (cm) and bulk density (g cm-3) of soil samples.".
L637, "P, I, and W were rainfall (mm)" modified to "P, I, and W are rainfall (mm)".
- L26 Unclear meaning of the symbol “~”; it is used to express “about, approximately”. For the range of values use the standard symbol (dash —), here and elsewhere in the text.
Response: The author checked the entire manuscript and replaced the symbol representing the range of values from "~" to "—".
- L26-27 Use the same variant label, it's confusing (W1200 x I1200). Also, the designation of the variants with a four-digit number is confusing in connection with the N rates doses (I1800N60). What to use W0, W1, and W2?
Response: The author checked the whole manuscript and unified it with the same variant label, replacing the label like W1200 with I1200. The author used the designation of the variants with a four-digit number because the experimental treatment is relatively complex, and the label like I1800N60 is more intuitive to describe the experimental treatment, enhance the readability of the article, and facilitate the reader's understanding.
- L25-27 Try to simplify the sentence to improve readability. Simply, yields increased due to the combined application of irrigation and N in both years; why describe again "which remained at a high level....".
Response: The author simplified "The higher GY were obtained by the interaction of irrigation and N fertilizer, with an increase rate ranging from 15.84%—35.40%, which remained at a high level under I1200N60, I1800N60 and I1800N120 treatments in both years" to "The higher GY were obtained by the interaction of irrigation and N fertilizer, with an increase rate ranging from 15.84%—35.40%".
- L27-29 Again unclear, confusing sentence. What are we comparing with what?
Response: Thanks to the reviewers for their valuable comments.
L27-29, "And among the increased yield of oilseed flax, 39.70%—78.06%, 14.49%—54.11% and -10.6%—24.93% were contributed by the application of irrigation and nitrogen, and interaction of irrigation and nitrogen (I×N), respectively.".
This sentence is related to the description of the contribution of each factor (irrigation (I), nitrogen (N), and interaction of irrigation and nitrogen (I×N)) to the yield increase of oilseed flax (Table 6), and there is no comparison between treatments. The range of values is obtained from the contribution values of different factors to the yield in the two growing seasons in Table 6.
- L32 All abbreviations must be explained.
Response: Thanks to the reviewers for their valuable comments.
The authors indicated the full name of all abbreviations. All abbreviations are explained when they first appear in the manuscript, and the following are added: SS is the abbreviation for soluble sugar, and ST is the abbreviation for starch (L32).
- L33 Improve syntax, grammar “….GY increased of oilseed flax….” L36-37 “…GY increased of oilseed flax…”
Response: Thanks to the reviewers for their valuable comments.
L33-35, The authors modified "The structural equation model showed that the GY increased of oilseed flax by irrigation and N were mainly due to the varying degrees of increase in DMA, EC, and TKW." to "The structural equation model showed that the key factors to increasing the GY of oilseed flax by irrigation and nitrogen fertilization were the differential increase in DMA, EC, and TKW.".
L36-38, The authors modified "In conclusion, irrigation and nitrogen application and their coupling effects increased EC and TKW by increasing DMA and NSC synthesis in flax organs, ultimately achieving an increase in oilseed flax." to "The increase in EC and TKW were attributed to the promotion of DMA and NSC synthesis in oilseed flax organs by irrigation, nitrogen fertilization and their coupling effects.".
- L38-40 Is it possible to recommend specific doses of irrigation and fertilization based on a 2-year trial at one site on small 10 m2 plots?
Response: Thanks to the reviewers for their valuable comments.
The author believes that, on the basis of the two-year trial carried out on a small plot of 10 m2 plots, it is possible to provide a theoretical basis for recommending irrigation and fertilizer doses for agricultural flax production in the trial area and similar areas, based on the following points:
(1) According to the geographical characteristics of the test area, this test field is a terraced field, which is usually narrow in width and irregular in shape. Therefore, this situation limited the area of each plot tested.
(2) In previous studies, perhaps due to some external factors or other considerations, some experimental plots have an area of 2×4 m (Niknejhad et al., 2013)and plot size was 13.5 m2 (Wang et al., 2023). These studies all provide theoretical basis for high crop yields, provides useful information for optimising water and nitrogen management decisions for crop in semi-arid climatic regions, and contributes to the promotion of sustainable development of crop production and water-efficient agriculture. In addition, experiments on organic fertilizers (Ma et al., 2023). The area of a single community is also 2×4 m. This study involved the impact of organic fertilizers on the growth and yield formation of flax, providing scientific basis for high yield of flax.
Niknejhad, Y., Daneshian, J., Rad, A., Pirdashti, H. and Arzanesh, M.H., 2013. Effect of plant growth promoting rhizobacteria (PGPR) on leaf area duration (LAD) dynamics of rice (Oryza sativa L.) plants under nitrogen and water limited conditions. Res. Crops, 14: 345-349.
Wang, X., Xiang, Y., Guo, J., Tang, Z., Zhao, S., Wang, H., Li, Z. and Zhang, F., 2023. Coupling effect analysis of drip irrigation and mixed slow-release nitrogen fertilizer on yield and physiological characteristics of winter wheat in Guanzhong area. Field Crops Research, 302: 109103.
Ma, X., Gao, Y., Wu, B., Ma, X., Wang, Y., Yan, B., Cui, Z., Wen, M., Zhang, X. and Wang, H., 2023. Organic manure significantly promotes the growth of oilseed flax and improves its grain yield in dry areas of the Loess Plateau of China. Agronomy, 13(9): 2304.
Introduction
- L113 Check grammar “oil seedrape” (either oilseed rape or rapeseed”
Response: Thanks to the reviewers for their valuable comments.
The author changed "oil seedrape" to "oilseed rape"。
Results
- L147 The description does not agree with the data – interaction was not significant in six of 10 cases.
Response: Thanks to the reviewers for their valuable comments.
L152-153, the author modified "Irrigation (I), nitrogen (N) and I × N had significant effects on DMA (Table 1)." to "Nitrogen (N) had a significant effect on the accumulation of DMA, but irrigation (I) and I × N had significant effects on the DMA of some flax growth stages in both years (Table 1).".
- 4 Units of yield component and yield are missing
Response: Thanks to the reviewers for their valuable comments.
The author has added the units of yield component and yield in Table.4. As follows, EC (number plant-1); GN (number capsule-1); TKW (g); GY (kg ha-1).
- 5 Add the explanation of (probably) 95% confidence interval ranges
Response: Thanks to the reviewers for their valuable comments.
In Fig.5, "The confidence interval is 95%, which means that there is a 95% probability that the true value of a parameter will fall within the interval of the measurement results." was added.
- 5 Add to the caption that only significant differences are shown (better to show all differences and indicate significant ones)
Response: Thanks to the reviewers for their valuable comments.
Changes indicating the increase in percentage of current nitrogen/irrigation application level over previous level. Different letters in the same column means the significant difference between treatments at p < 0.05 level.
The authors showed all differences in Tab.5.
In Tab.5, all differences are marked with red boxes. No significance analysis was performed between "Changes". Changes indicating the increase in percentage of current nitrogen/irrigation application level over previous level. For example, a value of "Changes" of 5.4 means that the grain yield of N60 level is increased by 5.4% compared with that of N0 level. A value of "Changes" of 11.31 means that the I1200 level has increased by 11.31% compared with the I0, 4.83 means that the I1800 level is 4.83% higher than the I1200, and 16.69 means that the I1800 level has increased by 16.69% compared with the I0.
- 6 Consider explaining better how the contribution to yield increase is calculated (in Methods or here)
Response: Thanks to the reviewers for their valuable comments.
The authors added a formula for calculating the contribution to yield increase in Methods (L615-620).
N (IiNj) (Contribution to yield increase (%))=
I (IiNj) (Contribution to yield increase (%))=
Ii×Nj (Contribution to yield increase (%)) = 1-N (IiNj) (Contribution to yield increase (%))-I (IiNj) (Contribution to yield increase (%))
where, i is the irrigation level, j is the nitrogen application level.
For example, the calculation process of the contribution of each factor (irrigation (I), nitrogen (N), and interaction of irrigation and nitrogen (I×N)) to yield increase under I1200N60 treatment is as follows:
N (Contribution to yield increase (%))=
I (Contribution to yield increase (%))=
I×N (Contribution to yield increase (%)) = 1-N (Contribution to yield increase (%))-I (Contribution to yield increase (%)) (Since the increase in oilseed flax production is attributed to irrigation (I), nitrogen (N) and interaction of irrigation and nitrogen)
Discussion
- It is strange that you do not cite the results, eg WUE and others, from the same trial from previous years (eg. Cui et al. 2023).
Response: Thanks to the reviewers for their valuable comments.
In the discussion, the author quoted Cui et al. (2023) and discussed the related contents.
L514-518, "This study observed similar results to Cui et al. (2023), irrigation was the main factor in increasing oilseed flax production in dryland, but high-yield and high-resource utilization were contradictory. And although higher GY was obtained at I1800 level, higher WUE and WIUE were observed at I1200 level. " were added.
Methods
- L551 Add methods of available P and K determination and specify from what layer soil was sampled
Response: Thanks to the reviewers for their valuable comments.
The author added the measurement results of available P and K and the soil layer for measuring the basic nutrient content of the soil to the method.
L565, "available phosphorus 26.43 mg kg-1, available potassium 108.37 mg kg-1” and “Before planting oilseed, soil samples from 0-30 cm soil layer were collected to determine basic soil physicochemical properties. "。
- L547-8 Show at Fig.7 or add in the text average values for the growth season (April-August)
Response: Thanks to the reviewers for their valuable comments.
The author added the average rainfall and average air temperature from April to August in Fig.7. As shown in the picture:
Figure 7. The rainfall and temperature in growth seasons of oilseed flax.
- L560-1 Explain in Methods how you eliminated the edge effect, the water and N movement to adjacent plots.
Response: Thanks to the reviewers for their valuable comments.
The author explains the method to eliminate the edge effect (the water and N movement to adjacent plots) in the method.
L594-596, "To minimize disturbance of the experimental plots by adjacent plots due to the movement of water and nitrogen, plant samples were collected to avoid being collected at the edges of the plots." was added.
- L562 a bit clumsy sentence.
Response: Thanks to the reviewers for their valuable comments.
The author optimized this sentence to make it concise and clear.
L579, "Irrigation method was flood irrigation, irrigation water was introduced into each residential area through pipelines with diameter of 8 cm, and measured by the water meters.".
- L558-9 Specify the form of N fertilizer.
Response: Thanks to the reviewers for their valuable comments.
The author specified the form of N fertilizer.
L577-578, "Urea (46% N) was selected as the nitrogen fertilizer.".
- L570 Specify, how plants were sampled-cut at the soil surface?
Response: Thanks to the reviewers for their valuable comments.
The author explained that the plant sampling method is in "4.3.1. Dry matter accumulation.".
L598-600, "Ten oilseed flax plants were randomly selected from each oilseed flax plot, cutting the above-ground parts to measure above-ground DMA at seedling, budding, anthesis, kernel and maturity stages.".
- L581-2 Unclear, specify the harvest area for final yield determination. Explain in Methods how did you eliminate the edge effect?
Response: Thanks to the reviewers for their valuable comments.
The author specified the harvest area for final yield determination. The approach to minimizing the edge effect is explained in Methods.
L611-613, "The oilseed flax in each plot was harvested individually, and the GY was measured after air-drying and selection.".
L575-577, "To ensure even irrigation of each plot and minimize the edge effect, 50 cm width ridges were set up between each replicate and 40 cm width ridges were set up between each plot.".
- L587 The soil moisture has been determined in the way for hundred and more years, why did you cite the authors? The same applies for WUE and NDF. If you want to cite even a trivial method, you must cite the original work not study where the method was used recently.
Response: Thanks to the reviewers for their valuable comments.
The authors deleted the cited authors and literature. Result of modification can be found at 4.3.5 (L621-635).
- L599 Explain in Methods or in Discussion why drainage could not occur despite a very high single dose of irrigation - you did not provide any information about the water capacity of the soil layers and other factors that affect drainage. Without it, WUE cannot be reliably calculated.
Response: Thanks to the reviewers for their valuable comments.
The field experiments were conducted in 2019-2020 at Dingxi Academy of Agricultural Science (34.26°N, 103.52°E and altitude 2040 m). The experimental area is a typical semi-arid area with an average annual rainfall of about 380 mm, an average annual temperature of 6.3 °C and 2453 hours of sunshine per year. Although the two irrigation levels set in the experiment were 1200 m3 ha-1 and 1800 m3 ha-1, respectively, and the irrigation was divided into two periods on average, the evaporation in the experimental area was 1526 mm. Due to the special climate of this experimental area, drainage will not take place.
This study focused on the reasons why the above-ground parts of flax caused its yield increase, i.e. the carbon allocation strategies of oilseed flax and their relationship with yield and yield components under different irrigation and N application management, especially under different climatic conditions. In addition, this study is expected to screen irrigation and fertilization strategies suitable for experimental and similar areas based on the dual objectives of achieving high yield and efficient resource use. Therefore, the water capacity of the soil layers and other factors affecting drainage were not overly considered in this study.
There has been a lot of work and researches on the effects of irrigation and nitrogen fertilizer strategies for different crops on soil water (Dehghanisanij et al., 2020; Gao et al., 2023; Wang et al., 2015; Wan et al., 2022; Li et al., 2022; Sangha et al., 2023), and valuable scientific results and progress have been obtained. For example, increasing the amount of irrigation can increase root zone soil available water of crops, and increasing nitrogen application could significantly enhance nitrogen content and crop water consumption in root zone soils. However, low irrigation amount was detrimental to the function of fertilizer. Previous studies about the effects of irrigation and nitrogen fertilizer of oilseed flax were mainly focused on the accumulation and transportation of nitrogen of oilseed flax, as well as the importance of temporal and spatial distribution of soil water and water consumption characteristics on yield and yield formation (Cui et al., 2016; Cui et al., 2020; Yan et al., 2017; Wu et al., 2017; Cui et al., 2015). It can be seen that a large number of studies have explained the causes of crop yield increase from the perspective of soil, but few studies have paid attention to the contribution and change of irrigation and nitrogen fertilizer and their coupling effects on crop yield increase. While this study paid more attention to the reason why the aboveground parts of flax caused its yield increase, that is, the carbon allocation strategies of oilseed flax and their relationship with yield and yield components under different irrigation and N application managements, especially under different climatic conditions (precipitation distribution and ambient temperature), therefore, the main goal of this study is to explain the reasons for the yield increase around the temporal and spatial distribution of NSC in the aboveground organs of flax and its relationship with flax yield and yield components, to analyze the response of flax NSC to different climatic conditions (precipitation distribution and ambient temperature), to clarify the contribution of irrigation, N, and their interaction to the increase of GY of oilseed flax, and, combined with water and nitrogen production efficiency, to propose an appropriate irrigation and N application strategy for oilseed flax in semi-arid areas of Northwest China.
In addition, the author does not have the right to use the relevant data of the water capacity of the soil layers, and can only calculate water use efficiency based on these relevant data.
The references are as follows:
Dehghanisanij, H. and Kouhi, N. (2020). Interactive effects of nitrogen and drip irrigation rates on root development of corn (Zea Mays L.) and residual soil moisture. Gesunde Pflanzen, 72(4), 335-349. doi: 10.1007/s10343-020-00516-4.
Gao, R., Pan, Z., Zhang, J., Chen, X., Qi, Y., Zhang, Z., et al. (2023). Optimal cooperative application solutions of irrigation and nitrogen fertilization for high crop yield and friendly environment in the semi-arid region of North China. Agricultural Water Management, 283, 108326. doi: 10.1016/j.agwat.2023.108326.
Wang, X., Shi, Y., Guo, Z., Zhang, Y. and Yu, Z. (2015). Water use and soil nitrate nitrogen changes under supplemental irrigation with nitrogen application rate in wheat field. Field Crops Research, 183, 117-125. doi: 10.1016/j.fcr.2015.07.021.
Wan, F., Wu, W., Liao, R. and Wang, Y. (2022). Spatiotemporal Distribution of Water and Nitrogen in Border Irrigation and Its Relationship with Root Absorption Properties. Water, 14(8), 1253. doi: 10.3390/w14081253.
Li, Y., Huang, G., Chen, Z., Xiong, Y., Huang, Q., Xu, X. and Huo, Z. (2022). Effects of irrigation and fertilization on grain yield, water and nitrogen dynamics and their use efficiency of spring wheat farmland in an arid agricultural watershed of Northwest China. Agricultural Water Management, 260, 107277. doi: 10.1016/j.agwat.2021.107277.
Sangha, L., Shortridge, J. and Frame, W. (2023). The impact of nitrogen treatment and short-term weather forecast data in irrigation scheduling of corn and cotton on water and nutrient use efficiency in humid climates. Agricultural Water Management, 283, 108314. doi: 10.1016/j.agwat.2023.108314.
Cui Z.J., Liu D., Wu B., Yan B., Ma J., Zhao B.Q., Gao Y.H., and Niu J.Y. (2020). Effects of water and nitrogen coupling on grain yield formation and nitrogen accumulation, transportation of oil flax in dryland. Chinese Journal of Applied Ecology. 31(3): 909-918. doi: 10.13287/j.1001-9332.202003.028.
Yan P., Cui H.Y., Fang Z.S., Niu J.Y., and Gao Y.H. (2017). Effects of Supplemental Irrigation on Soil Moisture and Grain Yield of Linum usitatissimum Linn. Research of Soil and Water Conservation. 24(1), 328-333+341. doi: 10.13869/j.cnki.rswc.2017.01.044.
Wu B., Gao Y.H., Gao Z.N., Yan B., Zhang Z.K., Cui Z.J. (2017). Effects of Fertilization on Water Consumption Characteristics and Grain Yield of Linum usitatissimum L. in Dry Land. Research of Soil and Water Conservation. 24(3): 188-193. doi: 10.13869/j.cnki.rswc.2017.03.034.
Cui H.Y., Hu F.L., Fang Z.S., and Niu J.Y. (2015). Effects of Irrigation Amount and Stage on Water Consumption Characteristics and Grain Yield of Oil flax. Journal of Nuclear Agricultural Sciences. 29(04): 812-819. doi: 10.11869/j.issn.100-8551.2015.04.0812.
- Conclusion is rather weak, try to improve it, and do not only repeat the results.
Response: Thanks to the reviewers for their valuable comments. We have rewritten the conclusion.
L659-675, "In this region, irrigation and nitrogen application can regulate EC and TKW by promoting SS and ST synthesis in oilseed flax, further increasing GY, but this positive impact was limited with increasing irrigation and N application amount. In addition, drought and low temperature induced SS and ST synthesis to resist unfavorable environment, respectively. Water deficit is the main factor limiting the yield increase of oilseed flax in semi-arid areas. The contribution of N to the increased GY was restricted by irrigation, while the contribution of irrigation to the increased GY was less affected by N. The irrigation application amount of 1200 m3 ha-1 combined with the N fertilization rate of 60 kg ha-1 obtained higher grain yield and NPFP, as well as the highest WIUE, which was a suitable field management for oilseed flax production in semi-arid areas and similar areas.".

Reviewer 2 Report
Comments and Suggestions for Authors
Dear Editor,
This manuscript deals with a very important topic for the development of agribusiness in semi-arid regions. The most efficient use of water and nitrogen fertilization aligns with some of the sustainable development goals of the 2030 Agenda. The combination of irrigation depth x nitrogen fertilizer dose positively influenced some quality attributes of flax. In general:
- The manuscript is very well organized and meets the scope of the Plants journal;
- Qualified references were used in the manuscript;
- Abstract, Introduction, Results, Discussion, Methodology and Conclusion are adequate and well structured;
- Figures and Tables with excellent visual quality;
- The conclusions respond to the objective of the manuscript.
To further improve the manuscript, I have only two suggestions:
1) Replace the word precipitation with rainfall throughout the text, if possible; and
2) In the methodology, if possible, provide some information on the quality (pH, electrical conductivity, sodium and others) and origin of the water used for irrigation.
Author Response
Dear Editors and Reviewers:
Thank you for your letter and for the reviewers’ comments concerning our manuscript entitled “Irrigation, nitrogen fertilization and climatic conditions mediated the carbon allocation strategy and yield of oilseed flax in dryland” (ID: 3183177). Those comments are all valuable and very helpful for revising and improving our paper, as well as the important guiding significance to our researches. We have studied comments carefully and have made correction which we hope meet with approval. We have re-submitted our manuscript with tracked changes to highlight the revisions. The main corrections in the paper and the responds to the reviewer’s comments are as flowing:
- Replace the word precipitation with rainfall throughout the text, if possible.
Response: The author has checked the whole manuscript and replaced "precipitation" with "rainfall" in the whole manuscript.
(2) In the methodology, if possible, provide some information on the quality (pH, electrical conductivity, sodium and others) and origin of the water used for irrigation.
Response: The authors provided information on the source and quality of irrigation water.
L585-589, “Groundwater was used for oilseed flax irrigation during the two-year experiment. According to the public announcement of routine monitoring of water quality in centralized drinking water sources in county-level cities on the official website of Dingxi Municipal People's Government, the quality information of oilseed flax irrigation water in the past two years can be obtained. (Table 8)” was added.
Table 8 was added.
Table 8. Information on the quality of oilseed flax irrigation water
|
Year |
pH |
Sulfate |
Chloride |
Sulfide |
Nitrate |
Nitrite |
Ammonia nitrogen |
sodium |
|
|
|
2019 |
7.82 |
486.00 |
419.00 |
0.005 |
19.05 |
0.005 |
0.04 |
180.50 |
||
|
2020 |
8.21 |
392.00 |
330.00 |
0.005 |
19.6 |
0.005 |
0.04 |
174.00 |
Units are mg L-1, pH is dimensionless.

Round 2
Reviewer 1 Report
Comments and Suggestions for Authors
The authors accepted most of the suggestions, very carefully and in detail explained the changes in detail, edited and corrected the manuscript.
I have only several minor comments
Title: I thing, the singular (semi-arid area) should be used as the experiment was performed in one site.
Fig.5 There is no need to explain what 95% means, it was enough to state that "The confidence interval is 95%"
Methods.
L565 The available P and K determination methods have not been added, it is sufficient to state the name or extraction agent (e.g. Olesen, Mehlich3, ammonium acetate, etc.) of the method without inserting new citations in References.
Fig.7 It would be better to show average values for each month. If not, specify (in the graph or in the caption) that it is the average value of the total precipitation for a month, not for the entire period of 5 months.
Author Response
Dear Editors and Reviewers:
Thank you again for your letter and for the reviewers’ comments concerning our manuscript entitled “Irrigation, nitrogen fertilization and climatic conditions mediated the carbon allocation strategy and yield of oilseed flax in dryland” (ID: 3183177). Those comments are all valuable and very helpful for revising and improving our paper, as well as the important guiding significance to our researches. We have studied comments carefully and have made correction which we hope meet with approval. We have re-submitted our manuscript with tracked changes to highlight the revisions. The main corrections in the paper and the responds to the reviewer’s comments are as flowing:
- Title: the singular (semi-arid area) should be used as the experiment was performed in one site.
Response: Thanks to the reviewers for their valuable suggestions and comments. The authors modified " semi-arid areas " to "semi-arid area" in the title.
- 5 There is no need to explain what 95% means, it was enough to state that "The confidence interval is 95%".
Response: According to the reviewer's comments, the authors deleted the explanation for 95%, and reserved "The confidence interval is 95%.".
Methods.
- L565 The available P and K determination methods have not been added, it is sufficient to state the name or extraction agent (e.g. Olesen, Mehlich3, ammonium acetate, etc.) of the method without inserting new citations in References.
Response: The authors added the available P and K determination methods in L565-566. Available phosphorus was determined by NaHCO3 extraction–Mo–Sb anti spectrophotometric method. Available potassium was determined by NH4OAc extraction–flame photometric method.
- 7 It would be better to show average values for each month. If not, specify (in the graph or in the caption) that it is the average value of the total precipitation for a month, not for the entire period of 5 months.
Response: The authors added a note to Fig.7. L572-573, "The blue bar represents the total rainfall in a month, and the average rainfall represents the monthly rainfall averaged from the total rainfall from April to August.".
